# Bitter taste sensitivity in domestic dogs (*Canis familiaris*) and its relevance to bitter deterrents of ingestion

**Matthew Gibbs**[1,2]*, **Marcel Winnig**[3], **Irene Riva**[4], **Nicola Dunlop**[1], **Daniel Waller**[1], **Boris Klebansky**[5], **Darren W. Logan**[1], **Stephen J. Briddon**[2], **Nicholas D. Holliday**[2], **Scott J. McGrane**[1]

1 Waltham Petcare Science Institute, Waltham on the Wolds, Melton Mowbray, Leicestershire, United Kingdom, 2 School of Life Sciences, The Medical School, Queen's Medical Centre, University of Nottingham, Nottingham, United Kingdom, 3 AXXAM GmbH, Constance, Germany, 4 AXXAM SpA, IMAX Discovery Unit, Bresso, Milan, Italy, 5 BioPredict, Inc., Demarest, NJ, United States of America

* matthew.gibbs@effem.com

**Data Availability Statement:** All relevant data are within the paper and its Supporting Information files.

## Abstract

As the most favoured animal companion of humans, dogs occupy a unique place in society. Understanding the senses of the dog can bring benefits to both the dogs themselves and their owners. In the case of bitter taste, research may provide useful information on sensitivity to, and acceptance of, diets containing bitter tasting materials. It may also help to protect dogs from the accidental ingestion of toxic substances, as in some instances bitter tasting additives are used as deterrents to ingestion. In this study we examined the receptive range of dog bitter taste receptors (Tas2rs). We found that orthologous dog and human receptors do not always share the same receptive ranges using *in vitro* assays. One bitter chemical often used as a deterrent, denatonium benzoate, is only moderately active against dTas2r4, and is almost completely inactive against other dog Tas2rs, including dTas2r10, a highly sensitive receptor in humans. We substituted amino acids to create chimeric dog-human versions of the Tas2r10 receptor and found the ECL2 region partly determined denatonium sensitivity. We further confirmed the reduced sensitivity of dogs to this compound *in vivo*. A concentration of 100μM (44.7ppm) denatonium benzoate was effective as a deterrent to dog ingestion in a two-bottle choice test indicating higher concentrations may increase efficacy for dogs. These data can inform the choice and concentration of bitter deterrents added to toxic substances to help reduce the occurrence of accidental dog poisonings.

## Introduction

The sense of taste, or gustation, integrates with olfactory and somatosensory cues to give the overall perception of flavour once a food is accepted into the mouth. Taste perception is mediated through different groups of chemoreceptors expressed in the oral cavity [1,2]. These receptors have been extensively studied in humans and mice. Studies in other mammals are less common, particularly functional studies of cat and dog taste receptors [3–8], which is

**Funding:** All work in the manuscript was funded by Mars Petcare UK. We declare that M.G., N.D., D. W., D.W.L., and S.J.M. are all employees of Mars Petcare UK. M.G. contributed to the conception and design of the work, the acquisition, analysis, and interpretation of the data, and drafted and substantially revised the work. N.D. contributed to the acquisition, analysis, and interpretation of the data for the in vivo work and revised the draft of the work. D.W. contributed to the analysis and interpretation of the data for the in vivo work and revised the draft of the work. D.W.L. contributed to the conception, experimental design and substantially revised the work. S.J.M. contributed to the conception, experimental design, interpretation of the data for the in vitro and in vivo work, and substantially revised the work. The funder approved the study design, data collection and analysis, decision to publish, and preparation of the manuscript.

**Competing interests:** I have read the journal's policy and the authors of this manuscript have the following competing interests: M.G., D.W., D.W.L. N.D., and S.J.M. were all employees of Mars Petcare UK when the research was conducted. I.R. and M.W. are employees of Axxam S.p.A., Italy and Axxam GmbH, Germany. The study was funded by Mars Petcare UK. M.G. and S.J.M. are named inventors on a patent application on screening methods using canine Tas2r receptors for pet food products and compositions. This does not alter our adherence to PLOS ONE policies on sharing data and materials.

somewhat surprising given their popularity as companion animals and size of the petfood industry which has developed around them.

The perception of bitter taste is thought to be partly associated with rejection of a food and protects animals from ingesting potentially toxic substances [9,10]. However, this does not result in complete rejection of bitter tasting foods [11,12] and not all bitter tasting chemicals are toxic [13]. Bitter taste is mediated through a group of G protein-coupled receptors known as the Taste type 2 receptors (Tas2rs) [14–16] which are expressed in taste papillae on the tongue [14,16], other surfaces in the oral cavity, and other areas of the body [17–21]. In the case of cats, some studies on the functionality of the Tas2rs have been conducted [3,4], but at the time of writing the authors were not aware of any such studies for dogs.

Bitter taste in the domestic dog is of practical interest for three main reasons. First, pet dogs are often exclusively, or partially, fed a diet of commercially prepared pet food [22]. Any bitter taste from the raw materials used in the manufacture of pet food has the potential to negatively impact enjoyment. As sustainable sources of alternative and non-animal protein become more common in dog food, palatability challenges may become more prevalent [23–25]. Second, many veterinary medicines are delivered orally, but are often rejected by dogs due to their bitter taste. This can result in difficulties maintaining compliance, and negatively impact the dog's health [26]. Third, bitter-tasting chemicals are commonly used to deter pets (and children) from inappropriate or unwanted chewing and ingestion. Of particular importance here is preventing the ingestion of common household substances which are toxic [27,28]. Automotive antifreeze containing ethylene glycol is one important example. Ingestion of ethylene glycol at levels as low as 4.4mL/kg for dogs and 1.4mL/kg for cats can prove fatal [29]. Accidental exposure of pet dogs to such products is quite common [30]. Some mammals are less sensitive to commonly used bitter tastants than humans [31,32], but the molecular basis for these differences are unknown.

In order to understand the responses of dog Tas2rs, we deorphanised dog bitter receptors and characterised their receptive ranges using a heterologous cell-based model. We used an amino acid substitution approach to explore the molecular basis for certain differences in sensitivity when compared to human data and validated the difference seen *in vitro* through *in vivo* choice tests in different breeds of dog.

## Materials and methods

### Dog Tas2r sequences

Nucleotide and protein sequences for human and mouse Tas2rs were retrieved from Ensembl (www.ensembl.org) and used to perform blastn and tblastn searches on the dog genome (Can-Fam 3.1). Matching sequences were checked for an E-value (a value representing the likelihood of finding a similar match by chance) less than $1x10^{-5}$ and an open reading frame of >800bp. Sequences were then used as queries for searches against the non-redundant databases and were discarded if the closest match was not a Tas2r. In addition to this, sequences were compared to sequence variation data from the Dog Biomedical Variant Database Consortium (DBVDC), a database containing 648 dog genomes at the time of use [33]. Variants were assessed using the Variant Effect Predictor tool (VEP) [34] and only variants with medium or high impact were considered in further analysis. We used this data to ensure we had the most common version of the dog receptor sequences for use in our assay platform.

### Bitter compounds

Bitter test compounds were selected from the literature on human and mouse bitter taste responses [35–37]. A total of 48 compounds were initially tested with all dog Tas2rs in a pre-

screening experiment (n = 1, S1 Table). Only those inducing clear, specific responses (see Statistical analysis methods) in one or more dog bitter receptors were taken forward to full concentration-response testing (n = 2). Compounds were dissolved in assay buffer (130mM NaCl, 5mM KCl, 1mM MgCl$_2$, 2mM CaCl$_2$, 5mM NaHCO$_3$, and 20mM HEPES, pH 7.4) or in assay buffer with the addition of DMSO, not exceeding a final concentration of 0.6% (v/v), which was required to improve solubility for some of the compounds.

### *In vitro* receptor expression and calcium imaging analysis

Dog *Tas2r* sequences were generated by gene synthesis (Eurofins Genomics) and subcloned into the pcDNA5/FRT expression vector (ThermoFisher Scientific) downstream of the sequence for the first 45 amino acids of the rat somatostatin receptor [38]. Human embryonic kidney (HEK)-293T-PEAKrapid cells (ATCC; CRL-2828) stably expressing the chimeric G protein subunit Gα16i/o44 [39] were used for all experiments. In addition, cells expressing a Gα16/gust44 [40] chimeric G protein were used for comparative tests with denatonium benzoate (DB). Plasmid DNA containing the *Tas2r* sequence, or no receptor sequence (mock control), was transfected using Lipofectamine 2000 (ThermoFisher Scientific). Cells were tested 24 hours after transfection. Culture media was removed and replaced with assay buffer containing 2μM Cal520-AM calcium-sensitive dye (AAT Bioquest) and 2.5mM probenecid (ThermoFisher Scientific). Cells were incubated in the dark for 3 hours at room temperature, then washed with assay buffer immediately before data acquisition.

An initial screening of all dog Tas2rs with all compounds was performed to identify as many potential agonists as possible (n = 1). Compounds were tested near their maximum soluble concentrations and then at 1/10 and 1/100 dilutions. Compound-receptor combinations showing clear and specific activation (see Methods section for Statistical analysis) of the receptor were taken forward to full concentration-response testing. The responses of mock transfected cells to stimulation with adenosine 5'-triphosphate (ATP) (ThermoFisher Scientific) were used as a positive control for dye loading.

Two sets of concentration-response experiments were performed independently in different laboratories (n = 2). The first set of data were acquired using a FlexStation 3 (Molecular Devices) multimodal plate reader, while the second set used a FDSS/μCELL system (Hamamatsu). In both cases, the maximum change in fluorescence was divided by the baseline fluorescence before compound injection (ΔF/F$_0$). In both sets of experiments, two to four technical replicates were included for each data point and the data were combined for further analysis. All analyses were conducted with Excel 365 (Microsoft) and GraphPad Prism 8 (GraphPad Software).

### Amino acid substitution between hTAS2R10 and dTas2r10

In order to understand observed differences in the sensitivity of dog Tas2r10 (dTas2r10) to DB, we performed an amino acid swapping experiment where non-identical amino acids in the domains of dTas2r10 and human TAS2R10 (hTAS2R10) were substituted into one another. Amino acid sequences of human and dog Tas2r10 were aligned using Clustal Omega (1.2.4) [41]. Receptor structure was based on the published structure of hTAS2R10 [42]. We focused on amino acid differences that were within receptor regions typically involved in the formation of the binding pocket and interaction of hTAS2R10 with its ligands: transmembrane domain 3 (TM3), extracellular loop 2 (ECL2), transmembrane domain 5 (TM5), transmembrane domain 6 (TM6), extracellular loop 3 (ECL3) and transmembrane domain 7 (TM7) [36,42–44]. We generated five different dTas2r10/hTAS2R10 chimeras and one hTAS2R10/ dTas2r10 chimera (S2 Table). The chimeric receptors were tested with DB (active for

hTAS2R10, inactive for dTas2r10) and cucurbitacin B (active for both hTAS2R10 and dTas2r10) with the calcium imaging assay (n = 2).

### *In silico* modelling of Tas2r10 and denatonium benzoate

We used the AlphaFold [45,46] predicted structure for hTAS2R10, and also used this as the template for our homology model of dTas2r10 with the human ECL2 loop (Chimera1). The model was constructed using Modeler software, which included sequence alignments and the building of the homology model (Discovery Studio- BIOVIA, Dassault Systems). DB was manually docked into hTAS2R10 and Chimera1 utilising knowledge of the site-directed mutagenesis of hTAS2R10 [42]. Subsequently, the structures with docked DB were energy minimized [47].

### *In vivo* testing of dogs with denatonium benzoate

Several experiments were performed to confirm the sensitivity of dogs to DB. In our first pilot experiment, 10 miniature schnauzer dogs were offered a solution of 10μM DB in deionised water in a two-bottle choice test with plain deionised water. In subsequent experiments, a total of 76 dogs of three different breeds (miniature schnauzer (31), Labrador retriever (26), and cocker spaniel (19)), were tested using the same assay, but with a concentration of 100μM DB in deionised water.

All dogs were between 1–9 years old (mean 4.4 years, standard deviation ± 2.4 years) and neutered. Dogs were either bred at the Waltham Petcare Science Institute or obtained from approved breeders according to Home Office regulations. Equal numbers of males and females were included in all tests. The solutions were offered on two consecutive days for a period of 5 hours. During this time the dogs were housed individually with indoor and outdoor access and were not offered any other sources of water. At other times of the day the dogs had free access to water and were group housed with several pen mates. The positions of the solutions were swapped on the second day to account for positional bias. Dogs were fed a commercially available dry diet (Pedigree® Dry) with the amount offered calculated for each individual dog based on their current and ideal bodyweights. The food was offered during the 5-hour testing period. All animal studies were in alignment with the Mars Animal Research Policy (www.Mars.com) and the Animal (Scientific Procedures) Act 1986. On completion of the research, dogs were either retained for further studies or rehomed according to the Waltham standard homing policy. These studies follow the 3Rs approach to experimentation with animals in scientific research and the ARRIVE guidelines [48]. Studies were verbally approved by the Waltham Animal Welfare and Ethical Review Board.

### Statistical analysis methods

For our pre-screen *in vitro* data each test, which consisted of receptor expressing cells and mock controls tested with three concentrations of test compound and a buffer control, were assessed using a two-way ANOVA with Sidak's multiple comparison test. A *p*-value of ≤0.05 for any single concentration was considered significant. In addition to this, the data was visually assessed for signs of autofluorescence produced by the test compound or signal saturation. Compounds producing clear, significant responses were tested in the full concentration-response experiment.

In the concentration-response experiment, thresholds of activation were assessed using a Student's t-test to compare receptor transfected cells and the mock control. A *p*-value ≤0.05 was considered significant. In addition to this, evidence for a concentration dependent

response (a signal increase over more than one concentration point without activation in the mock cell line) was required for a receptor-compound combination to be considered active.

Our *in vivo* test data using a two-bottle choice test was assessed using the Tukey post-hoc multiple comparison test. In each experiment, the differences in intake were calculated, both in terms of g, and in g/kg bodyweight of the animal. For both, linear mixed-effects models were fitted, with breed as the fixed effect (if required) and animal as the random effect. Dog age was also considered as a fixed effect in the linear mixed-effects model. For the model where Breed is also a fixed factor, age was included as both a crossed fixed factor and as a nested factor. Means and 95% confidence intervals for each group were produced, with differences from 0 assessed for significance using the *p*-values produced via Tukey's post-hoc multiple comparison test. Additionally, for the experiment with the three breeds (Labrador retrievers, miniature schnauzers and cocker spaniels) group-to-group comparisons of the variance were carried out via pairwise F-tests. For each of the pairwise comparisons *p*-values were calculated, with family-wise error rate controlled for by using the global *p*-value (calculated from the corresponding ANOVA including all 3 groups) as a base for any of the pairwise *p*-values. Analysis was conducted with RStudio v4.1.2. (www.rstudio.com).

## Results

### Dogs have 16 putatively functional Tas2rs

The number of Tas2r genes in dogs has been assessed several times using the available dog reference genome at the time [49–52]. We conducted an independent analysis to confirm the sequence of all putatively functional dog Tas2rs. Our analysis revealed 16 putatively functional dog Tas2r genes, which was consistent with previous studies that identified either 15 or 16 putatively functional receptor genes in dogs (S3 Table). The organisation of *Tas2rs* in the dog genome is similar to that found in humans. There are 2 main clusters of *Tas2r* genes on dog chromosomes 16 and 27, which contain genes orthologous to those found on human chromosomes 7 and 12, respectively [53]. In humans, *hTAS2R1* is found on chromosome 5, while in dogs d*Tas2r1* is found on chromosome 34. One difference in chromosomal arrangement between the two species appears for d*Tas2r2*, which is found on dog chromosome 14, whereas the human pseudogene is found on chromosome 7. In the expanded cluster of 8 genes previously referred to as the anthropoid cluster [53] dogs have only one gene, *dTas2r43*, indicating this expansion event happened after the divergence of the Boreoeutheria (S1 Fig).

In some cases, variation was observed between the sequences found in the reference genome and the variation data from the DBVDC. The levels of variation we observed were similar to those previously observed in human *TAS2R*s. Receptors had between 1–16 medium or high impact variants (assessed using the VEP [34]), a figure which is close to the 1–12 variants previously reported for humans [54]. The most common sequence found in the DBVDC data was used (S4 Table). The functional dog Tas2r repertoire does not have a complete 1:1 orthology relationship with the human or mouse repertoire (S1 Fig). For example, human TAS2R2, 12, 62 and 67 are pseudogenes, yet orthologous receptors appear to be functional in the dog.

Notably the dog ortholog of hTAS2R9 was not included in our assessment due to a 31 amino acid truncation at the C-terminal when compared to hTAS2R9. Based on the structure of hTAS2R9, this would include the entire C-terminal and a small part of TM7 [55]. We therefore treated dTas2r9 as a pseudogene despite it having an open reading frame of 280 amino acids.

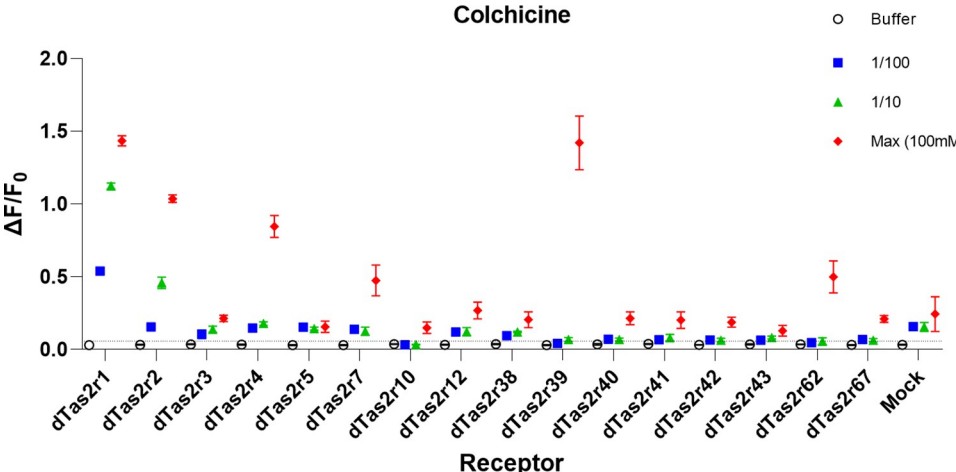

**Fig 1. Calcium responses of transfected HEK293T/Gα16i/o44 cells screened with three concentrations of Colchicine.** The receptors dTas2r1, 2, 3, 4, 5, 7, 38, and 39 were selected for testing in a full concentration-response experiment. Dog Tas2rs are on the x-axis, the y-axis shows the maximum change in fluorescence divided by the baseline fluorescence before compound injection ($\Delta F/F_0$).

## Identification of agonists for 7 dog Tas2rs

Pre-screening of the compounds allowed the rapid identification of potential agonists for dog Tas2rs. In a typical example (Fig 1), 100mM colchicine was tested with two 10-fold dilutions. Several receptors gave concentration-dependent responses while some others show responses only at the highest concentration. Only dTas2r1, 2, and 4 could be confirmed to have specific, concentration-dependent responses in subsequent concentration-response experiments (S2 Fig).

Forty-one (85%) compounds showed specific and significant activity (see Methods section for Statistical analysis) for at least one dog Tas2r in the pre-screen experiment or were selected based on their known activity with human taste receptors. These were tested in a concentration-response experiment (S5 Table). This approach resulted in the deorphanisation of 7 dog Tas2rs (44%) and the confirmation of 16 putatively bitter active compounds for dogs (Table 1, S2 Fig). The receptors with the greatest number of putative agonists were dTas2r1 and dTas2r4, both with eight. Multiple compounds activated three receptors, which was the highest total for any compound.

## The insensitivity of dog Tas2r10 to denatonium benzoate is partly related to differences in ECL2

Of particular interest were the responses of the dog Tas2rs to DB, due to its use as a chemical deterrent to ingestion. The most sensitive human receptor for DB is TAS2R47, which responds at concentrations as low as 30nM [35]. Dogs lack an orthologue of TAS2R47 but do have an orthologue for the next most sensitive human receptor, hTAS2R10. While dTas2r10 responded to another agonist of hTAS2R10, cucurbitacin B, we found it to be totally insensitive to the concentrations of DB tested here. Only dTas2r4 was responsive to DB, with an elevated activation threshold of 0.41mM (Table 1) compared to the human receptors [35]. To understand the reason hTAS2R10 and dTas2r10 have different sensitivities to DB, we performed an amino acid substitution experiment.

Amino acids previously shown to be involved in receptor-ligand interactions in hTAS2R10 [42] were compared to dTas2r10. Notably, all these amino acids were identical in human and

**Table 1. Confirmed agonists of dTas2rs.** A) A total of 7 dog Tas2rs were deorphanised with agonists showing specific, concentration-dependent responses. Threshold concentrations were the lowest concentration (mM) giving a significant (Student's t-test, $p \leq 0.05$) difference from the mock cell line. $EC_{50}$ values (mM) are given in parentheses where available (n.r. = no response, n.d. = not determined). B) Total discovered agonist counts for all dTas2rs.

| A | Compound | dTas2r1 | dTas2r2 | dTas2r4 | dTas2r5 | dTas2r10 | dTas2r12 | dTas2r41 | B | Receptor | Active compounds identified |
|---|---|---|---|---|---|---|---|---|---|---|---|
| | 1, 10-Phenanthroline | 1.25 (n.d.) | 1.10 (n.d.) | n.r. | 1.25 (n.d.) | n.r. | n.r. | n.r. | | dTas2r1 | 8 |
| | 6-Nitrosaccharin | 0.041 (n.d.) | 0.16 (0.28) | 0.31 (1.12) | n.r. | n.r. | n.r. | n.r. | | dTas2r2 | 6 |
| | (-)-α-Thujone | 0.025 (n.d.) | n.r. | n.r. | n.r. | n.r. | n.r. | n.r. | | dTas2r3 | 0 |
| | Aristolochic acid I | n.r. | 0.031 (0.037) | 0.039 (0.19) | n.r. | n.r. | n.r. | n.r. | | dTas2r4 | 8 |
| | Aurintricarboxylic acid | n.r. | 0.0031 (0.0034) | 0.0069 (n.d.) | 0.021 (n.d.) | n.r. | n.r. | n.r. | | dTas2r5 | 4 |
| | (-)-Camphor | 1.39 (n.d.) | n.r. | n.r. | n.r. | n.r. | n.r. | n.r. | | dTas2r7 | 0 |
| | Chlorhexidine | n.r. | n.r. | 0.0037 (n.d.) | n.r. | n.r. | n.r. | n.r. | | dTas2r10 | 1 |
| | Colchicine | 0.14 (n.d.) | 11.10 (n.d.) | 3.70 (n.d.) | n.r. | n.r. | n.r. | n.r. | | dTas2r12 | 1 |
| | Cucurbitacin B | n.r. | n.r. | n.r. | n.r. | 0.00069 (0.0024) | n.r. | n.r. | | dTas2r38 | 0 |
| | Denatonium benzoate | n.r. | n.r. | 0.41 (4.35) | n.r. | n.r. | n.r. | n.r. | | dTas2r39 | 0 |
| | Ethylpyrazine | 11.11 (n.d.) | n.r. | n.r. | n.r. | n.r. | n.r. | n.r. | | dTas2r40 | 0 |
| | Flavone | 0.00082 (n.d.) | n.r. | n.r. | n.r. | n.r. | 0.022 (n.d.) | n.r. | | dTas2r41 | 1 |
| | L-Menthol | 0.019 (n.d.) | n.r. | n.r. | n.r. | n.r. | n.r. | n.r. | | dTas2r42 | 0 |
| | Ofloxacin | n.r. | 0.69 (n.d.) | n.r. | n.r. | n.r. | n.r. | n.r. | | dTas2r43 | 0 |
| | Oxyphenonium bromide | n.r. | n.r. | 0.31 (4.76) | 3.70 (n.d.) | n.r. | n.r. | 11.11 (n.d.) | | dTas2r62 | 0 |
| | Sucralose | n.r. | n.r. | 1.23 (n.d.) | 33.33 (42.91) | n.r. | n.r. | n.r. | | dTas2r67 | 0 |

dog Tas2r10 and hence could not account for the observed difference in sensitivity towards DB. ECL2 proved to be critical to the receptor's sensitivity to DB. When the non-identical amino acids from the ECL2 domain of hTAS2R10 were substituted into dTas2r10 (Chimera1), some limited sensitivity to DB was observed, indicating the importance of the second extracellular loop in mediating sensitivity to DB (Fig 2). Consistent with this, substitution of the non-identical amino acids in the ECL2 domain of hTAS2R10 with those from dTas2r10 (Chimera6) resulted in a complete loss of sensitivity to DB. Independently substituting non-identical amino acids in four other domains of the dTas2r10 receptor with those from hTAS2R10 (Chimera2, 3, 4, and 5) did not confer any sensitivity (Fig 2). All the chimeric receptors retained sensitivity to cucurbitacin B, indicating they were functional. Chimeric receptors with a dog backbone did show some variation in sensitivity when compared to the native dTas2r10. However, the most noticeable difference in cucurbitacin B sensitivity was seen with Chimera6 which showed a large increase in $EC_{50}$ and a reduced maximum response when compared to the native hTAS2R10, indicating ECL2 does influence cucurbitacin B sensitivity, but not to the same extent as for DB (Fig 2).

Our *in silico* modelling data were consistent with our *in vitro* data. DB does not activate dTas2r10 due to differences in ECL2, notwithstanding the presence of tryptophan in dog ECL2 (the only ECL2 residue in contact with denatonium), the positioning of the tryptophan could be different in hTAS2R10, resulting in the loss of DB activity in dTas2r10. To compare residues in the hTAS2R10 sequence and the different dog-human chimeras all residues are given absolute numbers based on their sequence followed by Ballesteros-Weinstein numbering [56]. Denatonium has a quaternary ammonium cation and is positively charged. Our

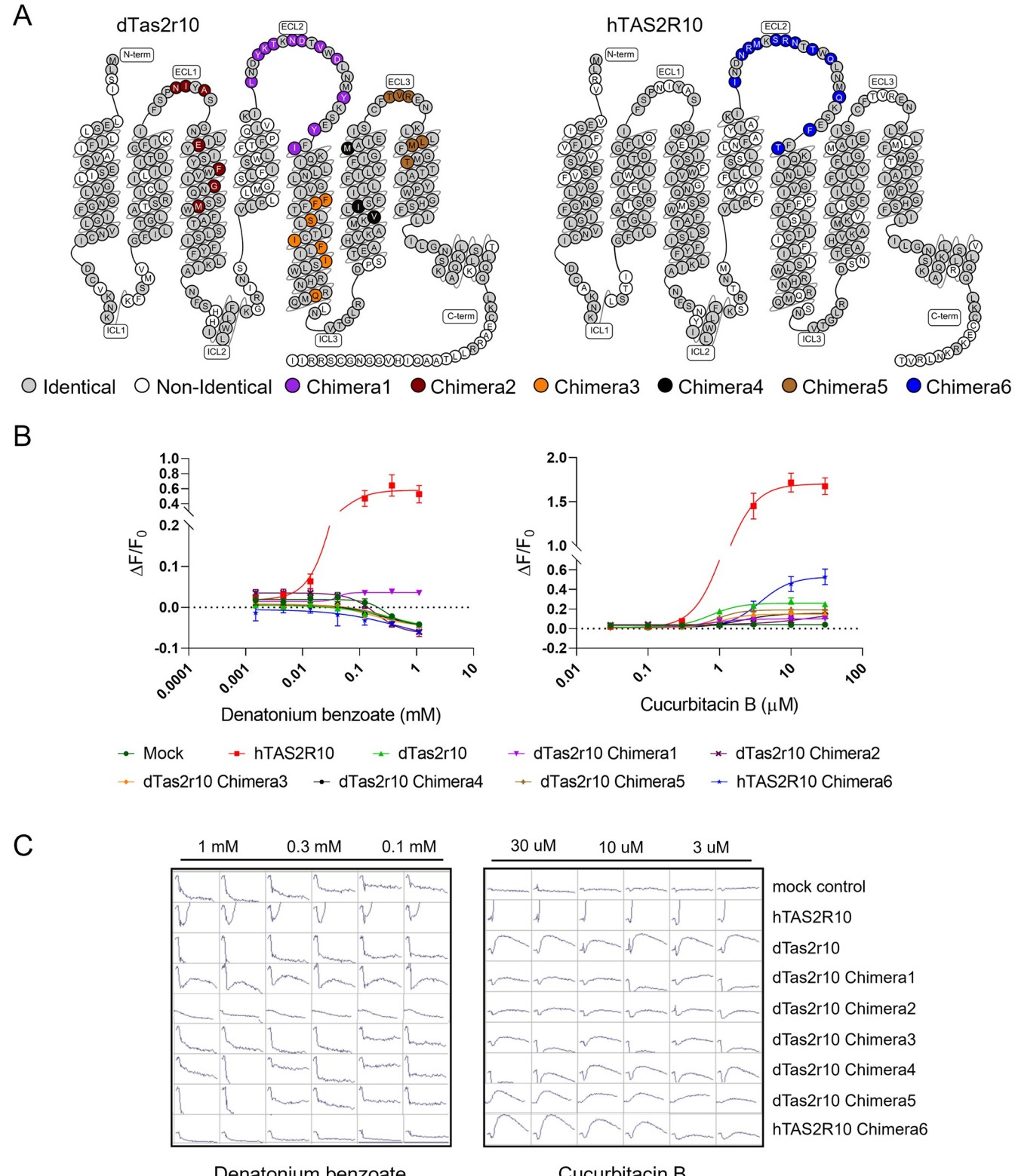

**Fig 2. Calcium imaging analysis of chimeric receptors.** A) Chimeras 1–5 were created with a dog Tas2r10 backbone and non-identical amino acids from different domains of hTAS2R10. Chimera6 consisted of the human hTAS2R10 backbone with the non-identical amino acids from dog Tas2r10 ECL2. B) Only native hTAS2R10 and dTas2r10 Chimera1, which contained the hTAS2R10 ECL2, responded to DB. Chimera6 (human TAS2R10 with dog ECL2) lost all

sensitivity to DB. All chimeric receptors retained their sensitivity to cucurbitacin B, although the response of Chimera6 was reduced compared to native hTAS2R10. DB and cucurbitacin B concentrations are plotted on a log scale on the x-axis, the y-axis shows the maximum change in fluorescence divided by the baseline fluorescence before compound injection ($\Delta F/F_0$). Colour scheme is common between panel A and panel B. C) Raw data is shown for all receptors with both DB and cucurbitacin B at the top three concentrations tested. Scale on the y-axis is -1000 to 2000 RFU with 70 seconds on the x-axis for all traces.

modeling of hTAS2R10 showed that there are two negatively charged residues, ASP64[2.60] and GLU246[6.58] that could make a salt bridge with the charged nitrogen of denatonium. The charged/pi interaction could also take place with the ring of TRP88[3.32]. SER85[3.29] and GLN175[5.40] could make a hydrogen bond to the amide group of the denatonium. The rest of the denatonium compound is hydrophobic, interacting with hydrophobic amino acids TRP88[3.32], LEU151[ex2], TRP162[ex2], LEU178[5.43], and MET263[7.39] (Fig 3).

For both hTAS2R10 and Chimera1 structures, the amino acids adjacent to DB (ASP64[2.60], GLN68[2.64], SER85[3.29], TRP88[3.32], LEU151[ex2], TRP162[ex2], TYR171[ex2], GLN175[5.40], LEU178[5.43], ASN179[5.44], TYR239[6.51], MET243[6.55], GLU246[6.58], PHE250[ex3], MET263[7.39], THR266[7.42], based on hTAS2R10 enumeration) were all identical, except for a MET243[6.55]ILE244[6.55] change, which changed methionine to the comparable hydrophobic amino acid isoleucine. This explains the activity of DB in Chimera1, but not the reduced sensitivity seen between Chimera1 and hTAS2R10. In Chimera1, there are other slight differences in amino acids that are proximate but not in direct contact with DB, glutamine changes to lysine (GLN68[2.64]LYS69[2.64]) and leucine changes to phenylalanine (LEU259[7.35]PHE260[7.35]). The variations MET243[6.55] ILE244[6.55], GLN68[2.64]LYS69[2.64], LEU259[7.35] PHE260[7.35], and other variations further from DB contribute to small changes in the shape of the Chimera1 active site pocket compared to hTAS2R10 and may be related to the lower activity of DB in Chimera1.

## Insensitivity of dog Tas2r10 to denatonium benzoate is replicated with Gα16/gust44 expressing cells

To further confirm our data for DB, we tested all four dog receptors orthologous to DB sensitive human receptors (dTas2r4, 10, 39, and 43) in a full concentration-response experiment. The human receptor thresholds of activation were previously published as 300μM for

A                                              B

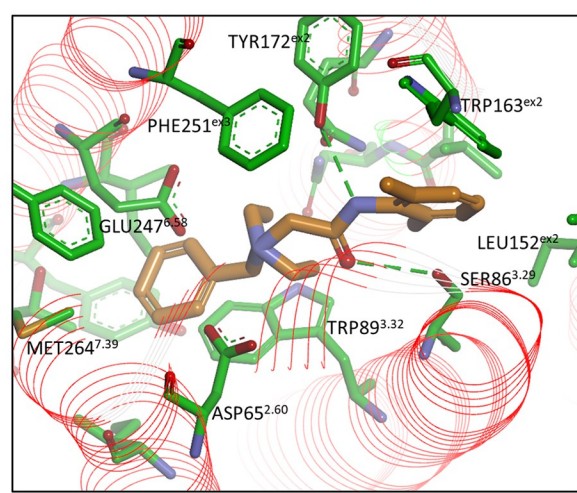

**Fig 3. *In silico* models of DB binding with hTAS2R10 and the dog/human chimeric receptor Chimera1.** A) hTAS2R10 residues are shown in blue, DB is in brown. B) Chimera1 residues are shown in green. Amino acids adjacent to DB are almost identical in the two receptors.

hTAS2R4, 3μM for hTAS2R10, 100μM for hTAS2R39 and 300μM for hTAS2R43 [35]. In addition to the Gα16/o44 cell line, we also used a cell line expressing the Gα16/gust44 G protein chimera, as was used to generate human TAS2R data. The dTas2r data from both cell lines were similar, with some exceptions. A small response was detected for dTas2r10 and dTas2r43 at the highest concentration of DB (10mM) with the Gα16/gust44 cell line only (Fig 4). While the response was not shown to be concentration-dependent, the data for 10mM DB were significantly different (Student's t-test, $p < 0.001$). These data suggest that the Gα16/gust44 chimera may be more sensitive with some receptor-compound combinations. However, the difference in performance of the cell lines is not enough to alter our conclusion that dogs lack a highly sensitive receptor for DB.

## Dogs show reduced sensitivity to denatonium benzoate when compared to humans

To ascertain whether insensitivity of dTas2r10 to DB resulted in a reduced perception of bitterness, we performed *in vivo* experiments with dogs and DB. A two-bottle choice test gives dogs a free choice between two solutions. In this case, the choice was always between water and the DB solution. The extent of a preference for drinking water over the DB solution, is interpreted as a measure of aversion due to bitterness. In the first pilot experiment, a concentration of 10μM DB failed to elicit a difference in preference in a panel of 10 miniature schnauzers ($p = 0.883$, Fig 5A). This concentration is well above the threshold of detection for humans and approaches concentrations previously show to reduce intake [27], suggesting dogs may indeed have a reduced sense of DB bitterness. In subsequent experiments, a higher concentration of 100μM DB did result in a difference in preference in three different breeds (cocker spaniels (n = 19) $p < 0.001$, Labrador retrievers (n = 26) $p < 0.001$, and miniature schnauzers (n = 31) $p < 0.001$, Fig 5B). In all three experiments the dogs preferred drinking water over DB, confirming it is probably perceived as bitter in dogs, albeit only when ingested at significantly higher concentrations than humans. Data were adjusted for bodyweight due to dog size variation both between and within breed, but similar results were calculated when DB intake was unadjusted for bodyweight (S3 Fig). We applied a pairwise F-test for equality of variance to all group-to-group comparisons. Labrador retrievers and miniature schnauzers were not found to have significantly different variances ($p = 0.225$). Cocker spaniels were significantly different from the other two breeds ($p = 0.031$), with a greater preference for water over DB, on average. Dog age was included as a fixed effect in the linear mixed-effects model and as both a crossed fixed factor and as a nested factor in the model where breed is also a fixed factor. For all tests age was not significant in any case ($p > 0.05$).

## Discussion

Bitter taste receptor repertoires vary in size considerably among mammals. Carnivores have previously been noted to have the smallest Tas2r families, with herbivores having repertoires of intermediate size and omnivores having the most Tas2rs [51]. This variation is thought to be driven, at least in part, by dietary specialisation [49,50,57,58]. Carnivores are unlikely to encounter a wide array of bitter compounds in their diet, while herbivores and omnivores encounter considerably more. Dogs are not obligate carnivores, but are opportunistic in their feeding, with fruit and vegetable matter contributing to the diet where available. This is also true of the domestic dog's relative, the gray wolf (*Canis lupus*) [59]. A Tas2r repertoire size of 16 is among the larger Tas2r gene families within the order *Carnivora* and is larger than that of obligate carnivores like the cat (*Felis catus*, 12 Tas2rs [4,51]), but appears to be consistent with the dog's semi-carnivorous nature.

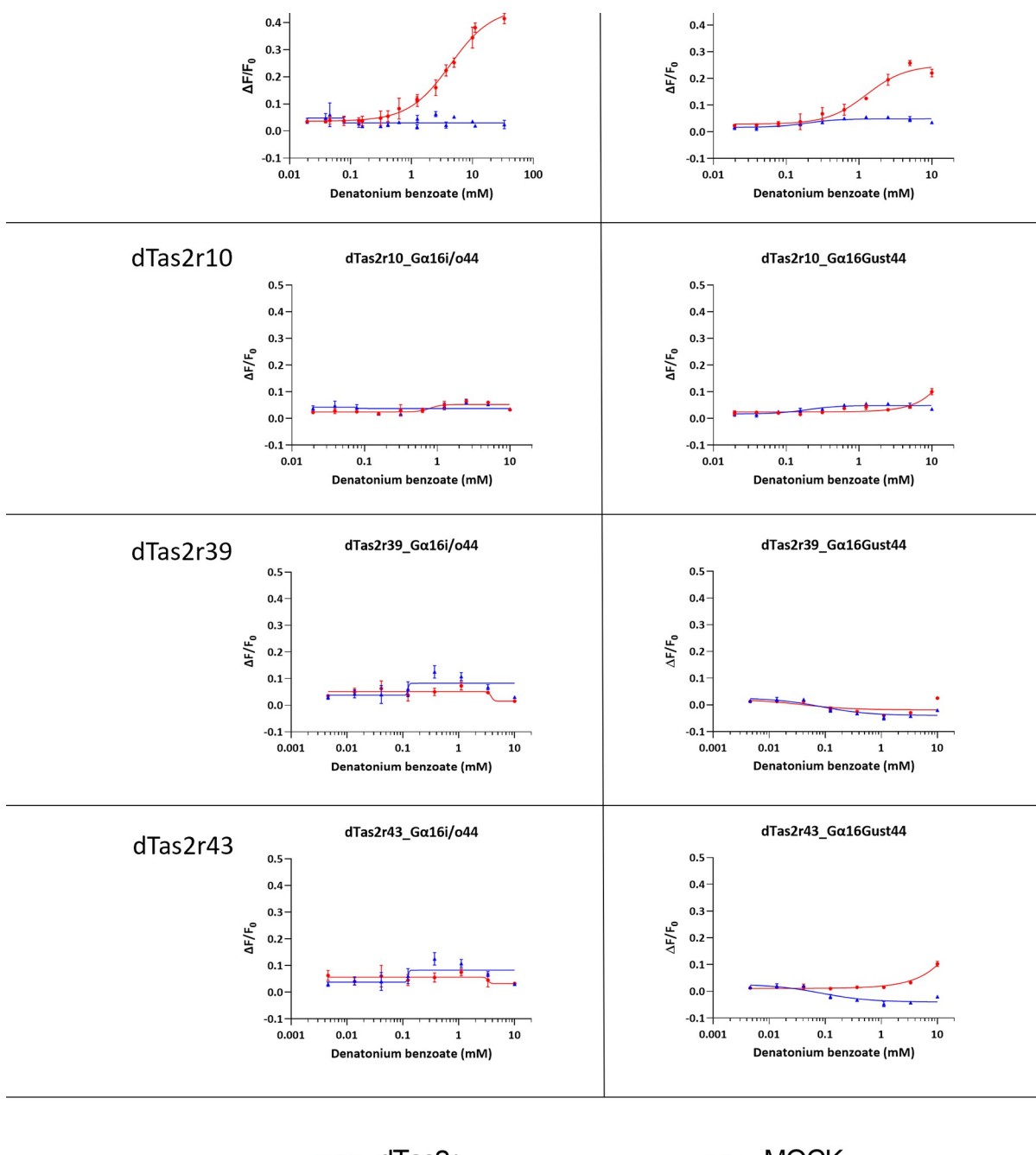

**Fig 4. Testing of DB with orthologous receptors in dogs and humans with two different recombinant cell lines.** In addition to the Gα16/o44 cell line, the Gα16/gust44 cell line was used. Results were similar with concentration-dependent responses only obtained with dTas2r4. When using the Gα16/gust44 cell line, small responses were observed with the highest concentration of DB only (10mM) for dTas2r10 and dTas2r43. These responses were significant (Student's t-test, $p < 0.001$). Compound concentration is plotted on a log scale on the x-axis, the y-axis shows the maximum change in fluorescence divided by the baseline fluorescence before compound injection ($\Delta F/F_0$). Data for dTas2r4 is taken from the confirmatory screening and thus appears in S2 Fig also.

In this study we expressed and tested all 16 dog Tas2rs in a heterologous cell-based assay. We utilised a G protein chimera previously shown to reliably couple to 20 of the 25 human bitter taste receptors [39]. Of the five-remaining human TAS2Rs, four are orphan receptors with

A

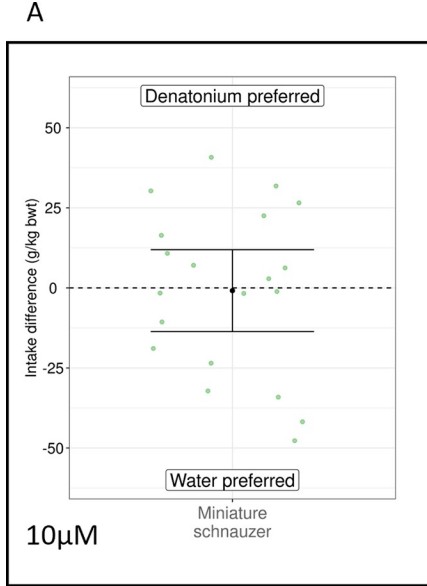

B

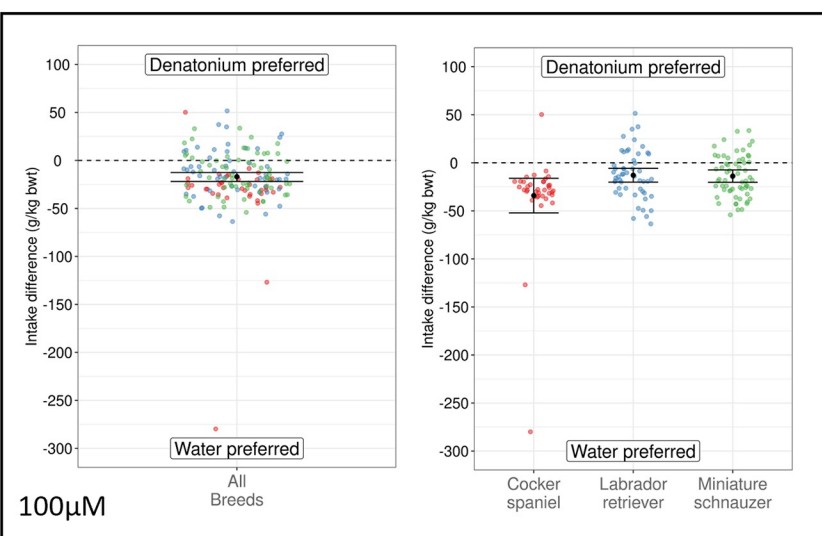

**Fig 5. Responses of dogs to DB at concentrations of 10μM and 100μM vs water.** Data are shown as intake difference in g/kg bodyweight. Data points show each individual exposure event. A) Ten miniature schnauzers were exposed to 10μM DB and water on two consecutive days, showing no significant difference in preference (mean = 0.866, 95%CI = -13.656 to 11.923, $p$ = 0.883). B) 76 dogs of three different breeds were exposed to 100μM DB on two consecutive days, showing a significant mean preference for water over DB, (mean = -16.949, 95%CI = -21.931 to -12.590, $p$ <0.001). When data from the three breeds are analysed independently, cocker spaniels (mean = -34.132, 95%CI = -54.020 and -14.244, $p$ <0.001), Labrador retrievers (mean = -12.955, 95%CI = -20.872 and -5.039, $p$ <0.001), and miniature schnauzers (mean = -13.828, 95%CI = -20.939 and -6.717, $p$ <0.001) all demonstrated a significant mean preference for water over DB. The Tukey post-hoc multiple comparison test was used to produce $p$-values.

no known agonist, which therefore cannot be shown to couple effectively. The fifth (TAS2R41) has only 2 known agonists [60].

We opted to use a pre-screening experiment in order to quickly identify candidate hit compounds, followed by further experiments testing full concentration-response ranges to confirm compound activity and assess receptor thresholds of activation and EC$_{50}$ values. A limitation of this approach is that some putative agonists may have been overlooked due to having a small concentration range over which the compound can be successfully tested. This is most often due to higher concentrations of the compound causing non-specific responses. In such cases the optimal test concentration may have fallen between the concentrations tested in the pre-screen. We therefore do not consider lack of a specific response in the pre-screen experiment as a negative result and only viewed the pre-screen data as a guide for compound selection for full concentration-response testing. Ultimately, 7 dog Tas2rs were successfully deorphanized, with agonists that showed concentration dependent responses.

Dog Tas2r1 and dTas2r4 showed the broadest activity in this study (eight agonists each), followed by dTas2r2 (six agonists), and dTas2r5 (four agonists). The agonists identified for the dog Tas2rs are not all shared by their human orthologues. In the case of dTas2r1, for example, 1, 10-phenanthroline, ethylpyrazine and colchicine are inactive against hTAS2R1 [35]. In general, the responses observed here showed that while orthologous receptors between dogs and humans do sometimes share common agonists, orthology is certainly not a reliable guide regarding receptor sensitivity to specific compounds. These data are consistent with comparisons between human and mouse Tas2r responses [53], which also show differences in receptive ranges between orthologous receptors.

We observed complete insensitivity of dTas2r10 to DB, in contrast to its human orthologue. Amino acid swapping experiments showed differences in ECL2 were at least partly the cause

of the insensitivity. ECL2 is widely recognised to play important roles in ligand binding and selectivity in many GPCRs [61,62], including the Tas2rs [63]. It is also implicated in maintaining the inactive conformation of some GPCRs [64]. The ECL2 regions of dTas2r10 and hTAS2R10 share 12 common residues and a further three have similar properties. The dTas2r10 ECL2 contains one additional residue and there are eight residues that differ between the two sequences. ECL2 is a highly flexible structure, and we could not use *in silico* modelling to show the impact of ECL2 on DB binding effectively. We did model the interactions of DB with the transmembrane binding site, which showed the majority of the interacting residues to be identical in hTAS2R10 and a chimeric receptor consisting of dTas2r10 with human ECL2. However, this chimeric receptor was only slightly sensitive to DB when compared to the native hTAS2R10, indicating some other variations between the two receptors play a role in their sensitivity. Differences in sensitivity were also observed with the triterpene cucurbitacin B. While dTas2r10 was activated by the compound the $EC_{50}$ was elevated and the maximum activation level was decreased when compared to hTAS2R10. Sensitivity was maintained with all dog-human Tas2r10 chimeras, although some variation in sensitivity between the different chimeras was observed. In particular, the human-dog chimera incorporating dog ECL2 showed a large reduction in maximum response and an increase in $EC_{50}$, indicating that the sequence of dog ECL2 may partly be the cause of the lower sensitivity observed with dTas2r10. Recently, the first ever crystal structure of a TAS2R receptor was published [65]. The authors identified differences in the position of ECL2 in a strychnine-bound TAS2R46-miniGs/gust complex when compared to the unbound form. Using dTas2r10 as a naturally occurring DB insensitive variant combined with mutagenesis of individual amino acids in ECL2 may provide additional information on the role of ECL2 in dTas2r10 DB sensitivity and on Tas2r activation mechanisms in general.

The compound library used here for the *in vitro* screening experiments consisted of 24 naturally occurring and 24 synthetic compounds (S1 Table). The compounds that were active against dog Tas2rs were also split equally, with eight natural and eight synthetic compounds stimulating at least one dog Tas2r. All of the eight naturally occurring dTas2r agonists were of plant origin, with some having known pharmacological activities. For example, colchicine occurs naturally in the autumn crocus (*Colchicum autumnale*), among other plants [66]. It is used to treat gout [67] and Behçet's disease [68], both inflammatory conditions. Some of the other active natural compounds are used in the food and flavour industry such as L-menthol and ethylpyrazine. The selection of synthetic active compounds also contained several with pharmacological and flavour activity, including, artificial sweeteners (sucralose), antimuscarinic drugs (oxyphenonium bromide) and antimicrobials (chlorhexidine, which is incorporated into some dog chews to help control periodontal disease [69]).

Of particular interest were the responses of dog Tas2rs to DB, which is often cited as the most bitter tasting chemical to humans [27,70]. This, and other properties like chemical stability and low toxicity, make it ideal for use as an additive to some toxic household products to deter ingestion by both humans and pets. However, not all species display the same sensitivity to the taste of this chemical. For example, rodents are less sensitive to DB than humans, a fact exploited in its use in increasing the selectivity of rodenticides [71]. We found only dTas2r4 to be sensitive to DB, while in humans eight Tas2rs are activated by this compound [35], with an active concentration as low as 30nM. Mice have five DB-sensitive receptors, but none had an active concentration lower than 100µM [53]. In our data the lowest significantly active concentration for dTas2r4 was greater, at around 411µM. At the next lowest concentration we tested (137µM) the responses were indistinguishable from the mock transfected cells.

To confirm the sensitivity of dogs to DB, we tested two concentrations of the compound in a series of experiments where dogs were given a choice of water or water containing DB. We

found that three different breeds of dogs rejected DB at 100µM. These data indicate that our *in vitro* assay lacked some sensitivity when compared with *in vivo* responses. Other published studies using similar heterologous cell-based assay systems have either reported similar thresholds when comparing *in vitro* and *in vivo* data for human receptors [72], or showed that *in vitro* thresholds were the same or lower when compared to *in vivo* thresholds in chickens [73]. It may be possible that our assessment of the number of functional Tas2rs in the dog genome is incomplete, or another receptor with greater sensitivity to DB remains to be discovered. The higher concentrations of DB used in our *in vivo* experiments could also result in other taste properties, such as astringency, that lead to rejection before the bitter taste became apparent.

Not all dog breeds behaved identically when offered solutions of DB. Cocker spaniels showed significantly different variance to both Labrador retrievers and miniature schnauzers ($p = 0.031$). This could indicate that some differences in DB sensitivity exist between dog breeds. Variation in the sequence of human TAS2Rs is known to play a role in sensitivity to some bitter compounds [74,75]. The same may be true for dogs, with differences disproportionately prevalent in some breeds due to the selective breeding that has shaped pedigree dog populations [76,77]. We saw comparable levels of genetic variation in dog Tas2rs when compared to humans. Average numbers of variants per gene were similar with 5.4 and 4.2 variants per gene respectively. Further work to assess the functional impact of these variants and assess their distribution among dog breeds would add to our understanding of bitter taste sensitivity in dogs. The dogs used in this study were all aged between 1–9 years old and we saw no relationship between DB sensitivity and age here. Further research over a more expanded age range of dogs (including juveniles and more senior dogs) may be needed to identify potential differences in sensitivity to DB. Age related differences in bitter taste sensitivity in both children [78] and the elderly [79,80] have been observed for humans, and further work to establish if this is also the case for dogs would be of interest, not only from the point of view of bitter taste but more generally in relation to flavour, food intake and bodyweight management in dogs, particularly seniors.

A concentration of 100µM DB (44.7ppm) falls within the upper range of DB concentrations previously proposed (30-50ppm) for use as a bitter deterrent in automotive antifreeze containing ethylene glycol. In the United States, this level was proposed in the Antifreeze Bittering Agent Act of 2005 which was not passed, but has since been adopted by all 50 U.S. states on a voluntary basis [81]. In the UK there is currently no legal requirement to add a bitterant to ethylene glycol containing products, but some manufacturers include one voluntarily. Although, on average, the dogs in our study did reject 100µM DB, there were some individual preference tests where the opposite was true, particularly for a few dogs in the Labrador retriever and miniature schnauzer groups. This suggests that the concentration of DB typically found in automotive antifreeze may not be sufficient for total rejection by some dogs. Further testing of higher DB concentrations with dogs would confirm whether that approach would prove an effective way of reducing accidental ingestion.

The requirements for a chemical to be useful as a bitterant are quite numerous. Any candidate must possess a widely perceived aversive taste, acceptable levels of stability, cost, toxicity, and environmental persistence, while not interfering with the function of the product. Of the other dog Tas2r agonists we discovered, cucurbitacin B was the most potent, with a detection threshold of 0.69µM and an $EC_{50}$ of 2.4µM against dTas2r10. Cucurbitacin B is also an agonist of hTAS2R10 [35], with a detection threshold of 0.01µM. In this case, the dog and human orthologues were similarly sensitive. Cucurbitacin B is a member of a family of compounds that occur naturally in many plants, notably the Cucurbitaceae family (pumpkins and gourds) and so is broadly available. However, cucurbitacin B has limited potential as an effective bitter deterrent due to its known toxicity in humans [82].

Taken together, we hope this work will inform the debate on the appropriate use of bitter deterrents to address the very real problem of accidental poisonings of pets.

## Supporting information

**S1 Fig. Phylogenetic tree of all intact human, mouse, and dog Tas2rs.** The tree was generated using the neighbour joining method with Jukes-Cantor protein distance measure and 1,000 bootstrap repetitions.
(DOCX)

**S2 Fig. All positive receptor-compound combinations.** Raw data is shown in each case for the highest concentration where a specific response was observed. Concentration-response curves show specific, concentration dependent responses. Where higher concentrations caused high activation in the mock transfects, these have been omitted for clarity. For all experiments n = 2 (error bars = SEM) with the exception of dTas2r2 with ofloxacin where n = 1 (error bars = SD). Compound concentration is plotted on a log scale on the x-axis, the y-axis shows the maximum change in fluorescence divided by the baseline fluorescence before compound injection ($\Delta F/F_0$).
(DOCX)

**S3 Fig. Responses of dogs to DB at concentrations of 10μM and 100μM vs plain water.** Data are shown as intake difference (g). Data points show each individual exposure event. A) Ten miniature schnauzers were exposed to 10μM DB and plain water on two consecutive days. A significant mean preference for plain water was not apparent (mean = -5.85, 95%CI = -111.89 to 100.19, $p$ = 0.905). B) 76 dogs of three different breeds were exposed to 100μM DB on two consecutive days. A significant mean preference for plain water was seen in the data for all 76 dogs (mean = -244.63, 95%CI = -316.83 to -174.35, $p$ <0.001). Data is also shown separately for the three breeds, cocker spaniels (mean = -355.34, 95%CI = -509.05 and -201.63, $p$ <0.001), Labrador retrievers (mean = -359.41, 95%CI = -561.43 and -157.39, $p$ <0.001), and miniature schnauzers (mean = -112.81, 95%CI = -168.62 and -56.99, $p$ <0.001). A significant mean preference for plain water was seen in all three breeds. Data were assessed using the Tukey post-hoc multiple comparison test.
(DOCX)

**S1 Table. Bitter compounds used for pre-screening dog Tas2rs.**
(DOCX)

**S2 Table. Dog and human Tas2r10 chimeric receptor sequences.** Substituted amino acids are highlighted in yellow with bold text.
(DOCX)

**S3 Table. Previously reported Tas2rs in the dog and those identified in this study.** Alternative or previous gene numbering shown in brackets (suffix p = pseudogene).
(DOCX)

**S4 Table. Dog Tas2r sequences identified and used in this study.**
(DOCX)

**S5 Table. Compounds selected for full concentration-response testing with dog Tas2rs based on pre-screen testing.**
(DOCX)

**S1 Dataset. Pre-screen_48_cmpds_all_dTas2rs.**
(XLSX)

**S2 Dataset. Confirmed_responses.**
(XLSX)

**S3 Dataset. Tas2r_chimeras.**
(XLSX)

**S4 Dataset. 10uM_DB_vs_Water.**
(XLSX)

**S5 Dataset. 100uM_DB_vs_Water.**
(XLSX)

**S6 Dataset. Chimera_Comparisson.**
(XLSX)

## Acknowledgments

The authors thank Timo Vennegeerts and Giuliana Piazza (Axxam SpA.), Dr. Jay Slack (Givaudan Flavours Corp.), and Sara Amundson (Humane Society Legislative Fund) for their contributions to the work. The authors also thank the Dog Biomedical Variant Database Consortium (Gus Aguirre, Catherine André, Danika Bannasch, Doreen Becker, Brian Davis, Cord Drögemüller, Kari Ekenstedt, Kiterie Faller, Oliver Forman, Steve Friedenberg, Eva Furrow, Urs Giger, Christophe Hitte, Marjo Hytönen, Vidhya Jagannathan, Tosso Leeb, Frode Lingaas, Hannes Lohi, Cathryn Mellersh, Jim Mickelson, Leonardo Murgiano, Anita Oberbauer, Sheila Schmutz, Jeffrey Schoenebeck, Kim Summers, Frank van Steenbeek, and Claire Wade) for sharing whole-genome sequencing and variant data from dogs. The authors gratefully acknowledge that part of this work was supported by a PhD studentship agreement between the Waltham Petcare Science Institute and The University of Nottingham.

## Author Contributions

**Conceptualization:** Matthew Gibbs, Marcel Winnig, Nicola Dunlop, Darren W. Logan, Stephen J. Briddon, Nicholas D. Holliday, Scott J. McGrane.

**Data curation:** Matthew Gibbs, Marcel Winnig, Irene Riva, Nicola Dunlop, Daniel Waller.

**Formal analysis:** Matthew Gibbs, Marcel Winnig, Daniel Waller, Boris Klebansky.

**Funding acquisition:** Darren W. Logan, Scott J. McGrane.

**Investigation:** Matthew Gibbs, Irene Riva, Boris Klebansky, Stephen J. Briddon, Nicholas D. Holliday.

**Methodology:** Matthew Gibbs, Marcel Winnig, Irene Riva, Nicola Dunlop, Daniel Waller, Stephen J. Briddon, Nicholas D. Holliday.

**Project administration:** Matthew Gibbs, Darren W. Logan, Stephen J. Briddon, Nicholas D. Holliday, Scott J. McGrane.

**Resources:** Darren W. Logan, Scott J. McGrane.

**Supervision:** Darren W. Logan, Stephen J. Briddon, Nicholas D. Holliday, Scott J. McGrane.

**Validation:** Matthew Gibbs, Irene Riva.

**Visualization:** Matthew Gibbs, Boris Klebansky.

**Writing – original draft:** Matthew Gibbs, Marcel Winnig, Daniel Waller, Boris Klebansky.

**Writing – review & editing:** Matthew Gibbs, Marcel Winnig, Irene Riva, Nicola Dunlop, Daniel Waller, Boris Klebansky, Darren W. Logan, Stephen J. Briddon, Nicholas D. Holliday, Scott J. McGrane.

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
