## [Decision Letter · Decision Letter 0]

23 Aug 2022

PONE-D-22-17431Bitter taste sensitivity in domestic dogs (*Canis familiaris*) and its relevance to bitter deterrents of ingestion.PLOS ONE

Dear Dr. Gibbs,

Thank you for submitting your manuscript to PLOS ONE. After careful consideration, we feel that it has merit but does not fully meet PLOS ONE’s publication criteria as it currently stands. Therefore, we invite you to submit a revised version of the manuscript that addresses the few points raised by Reviewer #2 during the review process.

We look forward to receiving your revised manuscript.

Kind regards,

Wolfgang Blenau

Academic Editor

PLOS ONE

Journal Requirements:

2. Please provide information in your manuscript text regarding the source of the animal subjects used in this study.

Please also state in your manuscript text the status of the animal subjects after the research (i.e. rehomed, adopted, housed for further research, euthanized). If any animals were sacrificed, please include information regarding the method of euthanasia and describe any efforts that were undertaken to reduce animal suffering.

 "All work in the manuscript was funded by Mars Petcare UK. We declare that M.G., N.D., D.W., D.W.L., and S.J.M. are all employees of Mars Petcare UK.

M.G. contributed to the conception and design of the work, the acquisition, analysis, and interpretation of the data, and drafted and substantially revised the work. N.D. contributed to the acquisition, analysis, and interpretation of the data for the in vivo work and revised the draft of the work. D.W. contributed to the analysis and interpretation of the data for the in vivo work and revised the draft of the work. D.W.L. contributed to the conception, experimental design and substantially revised the work. S.J.M. contributed to the conception, experimental design, interpretation of the data for the in vitro and in vivo work, and substantially revised the work."

    "I have read the journal's policy and the authors of this manuscript have the following competing interests: M.G., D.W., D.W.L. N.D., and S.J.M. were all employees of Mars Petcare UK when the research was conducted. I.R. and M.W. are employees of Axxam S.p.A., Italy and Axxam GmbH, Germany.  The study was funded by Mars Petcare UK. M.G. and S.J.M. are named inventors on a patent application on screening methods using canine Tas2r receptors for pet food products and compositions."

7. Please include your tables as part of your main manuscript and remove the individual files. Please note that supplementary tables (should remain/ be uploaded) as separate "supporting information" files

Reviewers' comments:

Reviewer's Responses to Questions

**Comments to the Author**

1. Is the manuscript technically sound, and do the data support the conclusions?

Reviewer #1: Yes

Reviewer #2: Partly

2. Has the statistical analysis been performed appropriately and rigorously? 

Reviewer #1: Yes

Reviewer #2: I Don't Know

3. Have the authors made all data underlying the findings in their manuscript fully available?

Reviewer #1: Yes

Reviewer #2: Yes

4. Is the manuscript presented in an intelligible fashion and written in standard English?

Reviewer #1: Yes

Reviewer #2: Yes

5. Review Comments to the Author

Reviewer #1: I think the paper is on an interesting issue and the approach is very well structured. Beside being useful about the appropriate use of bitter deterrents to avoid accidental poisonings of pets, could also be useful to understand the evolution of this taste modality in different mammals. Just a small comment: in the introduction Line 41, I suggest to eliminate PRIMARLY. In fact the discovery of extraoral T2Rs and of their functions (metabolic and in innate immunity), open the panorama about the role of bitter taste receptors, as well as of what drove their evolution.

For the rest, the paper is clear and smooth.

Reviewer #2: The authors used in-vitro and in-vivo tests to deorphanize several canine bitter taste receptors, and then focused on denatonium benzoate and comparison to human sensitivity. This is an important contribution.

Below are my main comments and concerns:

Abstract

“but not in every exposure” – not clear

The authors discuss dog-human orthologs. There should be some more in-depth discussion about assignment of dog to human orthologs, and comparison to the mice-human case https://www.sciencedirect.com/science/article/pii/S0021925820412761?via%3Dihub

The authors mention the dog genomes database that holds genomes from many dogs from different breeds, and use three different breeds in the behavioral assays. They should explicitly explore the breed or individual variations (if any) in the bitter taste receptors genes.

The age of the dogs varies dramatically (1-9). Distribution of ages should be shown and age should be taken into account in the results.

“ relationship between bitterness and toxicity has been questioned”. The authors cite a seminal paper from 1994(11), but are also encouraged to review an update from the same author https://link.springer.com/chapter/10.1007/164_2021_451 and a quantification paper https://pubmed.ncbi.nlm.nih.gov/29130618/

“All the chimeric receptors retained sensitivity to cucurbitacin B, indicating they were functional.” – but the activation levels in chimeras dropped dramatically, and in some of them there was also a large EC50 shift. These results should be explained and discussed in more detail.

“Insensitivity of dog Tas2r10 to denatonium benzoate is replicated with Gα16/gust44

expressing cells” – this should be also repeated for human Tas2R10 for proper comparison between the two types of in-vitro results. Or at least compared to results for human Tas2R10 activation by denatonium benzoate from the literature.

Discussion

“Other published studies using similar heterologous cell-based assay systems have reported similar thresholds when comparing in vitro and in vivo data for human receptors” –this may also be relevant to this discussion https://pubmed.ncbi.nlm.nih.gov/28513558/

6. PLOS authors have the option to publish the peer review history of their article (what does this mean?). If published, this will include your full peer review and any attached files.

Reviewer #1: **Yes: **Gabriella Morini

Reviewer #2: No

---

## [Author Response · Author response to Decision Letter 0]

6 Oct 2022

Response to reviewers:

The authors would like to thank the academic editor and reviewers for their time spent reviewing our paper and for the helpful feedback provided. Please see below a point-by-point summary of the changes made to the manuscript. We hope these adequately address the reviewer’s feedback, and we available in case of any further questions.

Journal Requirements: 

This has been addressed in the revised manuscript and supporting information.

2. Please provide information in your manuscript text regarding the source of the animal subjects used in this study.

Please also state in your manuscript text the status of the animal subjects after the research (i.e. rehomed, adopted, housed for further research, euthanized). If any animals were sacrificed, please include information regarding the method of euthanasia and describe any efforts that were undertaken to reduce animal suffering.

These points have been addressed in the revised manuscript.

 "All work in the manuscript was funded by Mars Petcare UK. We declare that M.G., N.D., D.W., D.W.L., and S.J.M. are all employees of Mars Petcare UK.

M.G. contributed to the conception and design of the work, the acquisition, analysis, and interpretation of the data, and drafted and substantially revised the work. N.D. contributed to the acquisition, analysis, and interpretation of the data for the in vivo work and revised the draft of the work. D.W. contributed to the analysis and interpretation of the data for the in vivo work and revised the draft of the work. D.W.L. contributed to the conception, experimental design and substantially revised the work. S.J.M. contributed to the conception, experimental design, interpretation of the data for the in vitro and in vivo work, and substantially revised the work."

This has been addressed in the amended cover letter.

 "I have read the journal's policy and the authors of this manuscript have the following competing interests: M.G., D.W., D.W.L. N.D., and S.J.M. were all employees of Mars Petcare UK when the research was conducted. I.R. and M.W. are employees of Axxam S.p.A., Italy and Axxam GmbH, Germany. The study was funded by Mars Petcare UK. M.G. and S.J.M. are named inventors on a patent application on screening methods using canine Tas2r receptors for pet food products and compositions."

This has been addressed in the amended cover letter.

The minimal data set has been supplied as supporting information.

This has been included in the manuscript.

7. Please include your tables as part of your main manuscript and remove the individual files. Please note that supplementary tables (should remain/ be uploaded) as separate "supporting information" files (as separate excel or pdf)

This has been addressed in the manuscript.

A number of new references have been added to the manuscript in response to the comments from Reviewer#2. 

Reviewer #1: I think the paper is on an interesting issue and the approach is very well structured. Beside being useful about the appropriate use of bitter deterrents to avoid accidental poisonings of pets, could also be useful to understand the evolution of this taste modality in different mammals. Just a small comment: in the introduction Line 41, I suggest to eliminate PRIMARLY. In fact the discovery of extraoral T2Rs and of their functions (metabolic and in innate immunity), open the panorama about the role of bitter taste receptors, as well as of what drove their evolution.

For the rest, the paper is clear and smooth. 

We thank Reviewer#1 for their feedback on the manuscript. The word PRIMARILY has been amended to PARTLY on Line 41.

Reviewer #2: The authors used in-vitro and in-vivo tests to deorphanize several canine bitter taste receptors, and then focused on denatonium benzoate and comparison to human sensitivity. This is an important contribution.

Below are my main comments and concerns:

Abstract

“but not in every exposure” – not clear

This text was intended to highlight the fact that in a small minority of cases when dogs were given a choice to drink water or a denatonium solution they would drink more of the denatonium solution. We now feel this is unnecessary in the Abstract and have removed it.

The authors discuss dog-human orthologs. There should be some more in-depth discussion about assignment of dog to human orthologs, and comparison to the mice-human case https://www.sciencedirect.com/science/article/pii/S0021925820412761?via%3Dihub

We have included further detail on the assignment of dog-human orthologs including a comparison of chromosomal locations on page 10. Not all dog Tas2rs are annotated in the dog reference genome. Those that are, have been assigned names equivalent to their closest matching human gene. We have followed this convention.

The authors mention the dog genomes database that holds genomes from many dogs from different breeds, and use three different breeds in the behavioral assays. They should explicitly explore the breed or individual variations (if any) in the bitter taste receptors genes. 

We completely agree and we have started some investigations on this subject. We hope to publish further on this in the future. We have added some details on the levels of variation we found in dog Tas2rs on page 11. We have started to assess the functional impact of some of these variants in our lab, but work is ongoing.

The age of the dogs varies dramatically (1-9). Distribution of ages should be shown and age should be taken into account in the results.

We have added more detail on the age range of the dogs and done further statistical analysis to assess the impact of age on the data. Details have been added in several places on pages 8, 9, 25 and 31. Age did not show any significant (p > 0.05) relationship with intake. This may be because we did not work with extremely young or old dogs, where differences in taste perception may be more apparent.

“ relationship between bitterness and toxicity has been questioned”. The authors cite a seminal paper from 1994(11), but are also encouraged to review an update from the same author https://link.springer.com/chapter/10.1007/164_2021_451 and a quantification paper https://pubmed.ncbi.nlm.nih.gov/29130618/

We agree that these references are important and have added them to the manuscript. We have also adjusted some of the text on page 3.

“All the chimeric receptors retained sensitivity to cucurbitacin B, indicating they were functional.” – but the activation levels in chimeras dropped dramatically, and in some of them there was also a large EC50 shift. These results should be explained and discussed in more detail. 

We agree and have added further discussion of these points on pages 15 and 22. In particular, the large increase in the EC50 for the chimeric receptor consisting of hTAS2R10 with dog ECL2 indicates that ECL2 plays a role in cucurbitacin B binding also.

“Insensitivity of dog Tas2r10 to denatonium benzoate is replicated with Gα16/gust44

expressing cells” – this should be also repeated for human Tas2R10 for proper comparison between the two types of in-vitro results. Or at least compared to results for human Tas2R10 activation by denatonium benzoate from the literature. 

We do not have data for human TAS2R10 and the Gα16/gust44 cell line currently. We have therefore referenced the published data for hTAS2R10 in more detail as suggested.

Discussion

“Other published studies using similar heterologous cell-based assay systems have reported similar thresholds when comparing in vitro and in vivo data for human receptors” –this may also be relevant to this discussion https://pubmed.ncbi.nlm.nih.gov/28513558/

We thank Reviewer#2 for highlighting this reference and have included it in our discussion.

We would like to explain two further updates to the manuscript to the journal editor and reviewers.

While in the process of reviewing our data after we had submitted the manuscript, we came to the conclusion that we should have included an additional dog-human chimera in our experiments. Previously we had not included a dog-human chimera incorporating variable residues in ECL1 and TM3. However, these regions have previously been implicated in TAS2R10 binding and we therefore conducted an additional experiment to include this chimera. The data were consistent with the other chimeras that did not involve ECL2, no sensitivity to DB was detected while low sensitivity to cucurbitacin B was maintained. The data have been incorporated into Fig 2 and the chimera numbering has been adjusted throughout the manuscript. 

We would also like to explain some minor changes to the EC50 values stated in Table 1. While we previously had two sets of experiments for this in some cases the concentration ranges used in the experiments differed. We decided to repeat some of these experiments to harmonise the concentration range which has resulted in some minor changes to the EC50 values stated in the table. This also resulted in some changes to Fig 4 and S2 Fig to reflect the updated data.

---

## [Decision Letter · Decision Letter 1]

1 Nov 2022

Bitter taste sensitivity in domestic dogs (Canis familiaris) and its relevance to bitter deterrents of ingestion.

PONE-D-22-17431R1

Dear Dr. Gibbs,

We’re pleased to inform you that your manuscript has been judged scientifically suitable for publication and will be formally accepted for publication once it meets all outstanding technical requirements.

Kind regards,

Wolfgang Blenau

Academic Editor

PLOS ONE

Additional Editor Comments (optional):

Reviewers' comments:

Reviewer's Responses to Questions

**Comments to the Author**

1. If the authors have adequately addressed your comments raised in a previous round of review and you feel that this manuscript is now acceptable for publication, you may indicate that here to bypass the “Comments to the Author” section, enter your conflict of interest statement in the “Confidential to Editor” section, and submit your "Accept" recommendation.

Reviewer #2: All comments have been addressed

2. Is the manuscript technically sound, and do the data support the conclusions?

Reviewer #2: Yes

3. Has the statistical analysis been performed appropriately and rigorously? 

Reviewer #2: I Don't Know

4. Have the authors made all data underlying the findings in their manuscript fully available?

Reviewer #2: Yes

5. Is the manuscript presented in an intelligible fashion and written in standard English?

Reviewer #2: Yes

6. Review Comments to the Author

Reviewer #2: Thanks for addressing reviewers comments. I believe that the paper, as well as all the data gathered in it, will be of interest to the scientific community.

7. PLOS authors have the option to publish the peer review history of their article (what does this mean?). If published, this will include your full peer review and any attached files.

Reviewer #2: No

---

## [Editor Report · Acceptance letter]

8 Nov 2022

PONE-D-22-17431R1 

Bitter taste sensitivity in domestic dogs (Canis familiaris) and its relevance to bitter deterrents of ingestion. 

Dear Dr. Gibbs:

I'm pleased to inform you that your manuscript has been deemed suitable for publication in PLOS ONE. Congratulations! Your manuscript is now with our production department. 

Kind regards, 

on behalf of

Dr. Wolfgang Blenau 

Academic Editor

PLOS ONE